# Italian Translation and Validation of the Readiness for Interprofessional Learning Scale (RIPLS) in an Undergraduate Healthcare Student Context

**DOI:** 10.3390/healthcare10091698

**Published:** 2022-09-05

**Authors:** Florian Spada, Rosario Caruso, Maddalena De Maria, Emiljan Karma, Aisel Oseku, Xhesika Pata, Emanuela Prendi, Gennaro Rocco, Ippolito Notarnicola, Alessandro Stievano

**Affiliations:** 1Department of Biomedical Sciences, Faculty of Medicine, Catholic University “Our Lady of Good Counsel”, 1000 Tirana, Albania; 2Department of Biomedicine and Prevention, University of Rome Tor Vergata, 00133 Rome, Italy; 3Department of Biomedical Sciences for Health, University of Milan, 20133 Milan, Italy; 4Health Professions Research and Development Unit, IRCCS Policlinico San Donato, 20097 San Donato Milanese, Italy; 5Research Centre on Developing Economies, Catholic University “Our Lady of Good Counsel”, 1000 Tirana, Albania; 6Faculty of Physical Activity and Recreation, Sports University of Tirana, 1001 Tirana, Albania; 7Centre for Excellence for Nursing Scholarship, OPI, 00173 Rome, Italy; 8Department of Clinical and Experimental Medicine, University of Messina, 98100 Messina, Italy

**Keywords:** collaborative teamwork, health professions, interprofessional education, psychometric evaluation, readiness for interprofessional learning scale

## Abstract

Interprofessional education requires that two or more professionals learn from and with each other to allow effective collaboration and improve health outcomes. Thus far, the interprofessional collaboration of healthcare students might be assessed using the Readiness for Interprofessional Learning Scale (RIPLS), which is currently not available in its Italian version. This study aimed to provide the intercultural adaptation of the RIPLS in Italian (I-RIPLS) and assess its validity and reliability. A two-phase validation study was performed in 2020, using a single-centre approach in students enrolled in the medical degree, physiotherapy, nursing, and dentistry courses at an Italian-speaking university in Albania. The first phase of the study determined the cross-cultural adaptation of the items by involving two translators who followed a forward and backward translation process. In the second phase, a sample of 414 students was enrolled. The preliminary corrected item-total correlations showed that five items did not show significant item-to-total correlations. Even if their deletion was not mandatory for generating a suitable correlation matrix for factor analysis, the advantages of keeping only items contributing to a more stable measurement with a shorter scale represented the rationale for removing items with non-significant item-to-total correlation from the correlation matrix before testing the dimensionality of the I-RIPLS with factor analysis. The answers from the first 50% of responders (*n* = 207) were used to determine the most plausible dimensionality of the I-RIPLS by employing an exploratory factor analysis (EFA), and the second 50% were used to cross-validate the most plausible dimensionality derived from EFA by employing confirmatory factor analysis (CFA) models. The most plausible dimensionality from EFA, by acknowledging the interpretation of the scree plot, the eigenvalues greater than 1, a parallel analysis, and the previous theoretical dimensions of the tool had two factors with adequate internal consistency. The CFA confirmed the two-factor solutions and the internal consistency for each domain. The I-RIPLS has 14 items with adequate evidence of validity and reliability. Future research should revise the tool for pursuing cross-cultural multigroup measurement invariance.

## 1. Introduction

The World Health Organization (WHO) defines interprofessional education (IPE) as when two or more professionals learn from and with each other to allow one effective collaboration and improve health outcomes [1,2]. The complexity of healthcare globally has been a driving force for IPE implementation at the international level [2,3]. IPE promotes a collaborative approach to developing healthcare students as future interprofessional team members [4,5]. Healthcare professionals-to-be should optimize their ability to participate in the new paradigm of healthcare delivery [6]. The recent literature sees IPE as an opportunity to change how future health workers are educated and as an occasion to take a step back and reconsider traditional healthcare practice [7,8].

In this context, teamwork is constituted by healthcare professionals of various specialities and, in every healthcare setting, stands as a crucial non-technical skill in ensuring good care to the patient [9]. The more the team members respect each other and know how to work collaboratively, the higher the quality and efficiency of patient care [3]. Therefore, it may be necessary to make structural changes within educational curricula, making patient-centred collaborative practice a responsibility of all healthcare programs [10]. Educators should conduct targeted student orientation based on data analysis and provide relevant support for IPE [11].

The students’ attitudes during formal education are recognized as the most important predictors of a successful implementation of interprofessional training to develop collaborative practice [12]. The opportunities for students to gain interprofessional experience help them to learn needed skills to become part of collaborative healthcare staff for practice [13]. A practice-ready collaborative workforce is a specific way to describe health professionals who have received adequate training in IPE [14]. Interprofessional instruction verifies when students of two or more professions learn from one another to implement effective collaboration and improve health outcomes. Once students understand how to work inter-professionally, they might be ready to enter workplace environments as members of the collaborative practice team [2].

Given the relevance of adequately assessing the readiness for interprofessional learning in students from several courses belonging to the healthcare professions, the Readiness for Interprofessional Learning Scale (RIPLS) has been developed and validated [15]. The RIPLS is one of the oldest and most widespread tools in assessing IPE, and it originally encompassed two dimensions and 19 items, even acknowledging that some further studies proposed a three-structure factorial model to explain which domains were measured by the RIPLS to assess IPE [15]. The RIPLS has proved helpful in university settings by enabling educators to assess students’ readiness to engage in IPE [16], and it stands as one of the most frequently applied tools for evaluating interprofessional education and learning activities [17].

Thus far, the RIPLS is a strategic tool for assessing IPE in healthcare university students and health professionals toward interprofessional learning, and for this reason, it is strategic to have a correct version of it in different languages and cultures [11]. In this regard, Table 1 shows an overview of the RIPLS characteristics and reliability coefficients from previous studies. Among the available versions, the one proposed by Reid [16] was translated and adopted by different countries, such as the versions in Portuguese (for Brazil) [18] and Arabic [19]. The version developed by McFadyen and colleagues [20] was one of the more widely adopted as it was translated into Chinese [11], French [21], German [22], Japanese [1], Serbian [12], Swedish [23], Turkish [24], and other languages. However, no Italian version had been prepared and validated before this study. For these reasons, this research aimed to provide the intercultural adaptation of the RIPLS developed by McFadyen and colleagues [20] into Italian (I-RIPLS) and assess its validity and reliability.

## 2. Materials and Methods

### 2.1. Design

This was a multiphase validation study. Firstly, a cross-cultural validation process was accomplished following well-established guidelines [33]. The first phase was the Italian translation and cross-cultural adaptation of the RIPLS (English version). The second phase was based on two rounds of cross-sectional data collection. More precisely, the first data collection round was required to assess the most plausible factor structure of the Italian version of the RIPLS using an Exploratory Factor Analysis (EFA), and the second round was aimed to cross-validate the most plausible factor structure (dimensionality) derived from the EFA by employing a Confirmatory Factor Analysis (CFA).

### 2.2. Description of the Version of RIPLS Adopted in This Study

The RIPLS is a widely used scale that measures the readiness of health care students for shared learning, and the most used version has been developed by McFadyen et al. [20]. The McFadyen et al. version [20] is a scale of 19 items, and it measures in the English-speaking population four domains: (1) Teamwork and collaboration; (2) Negative professional identity; (3) Positive professional identity; (4) Roles and responsibilities. Within the proposed model, the 4 domains were labelled as SS1, SS2, SS3 and SS4: SS1, Teamwork and collaboration (items 1–9); SS2, Negative professional identity (items 10–12); SS3, Positive professional identity (items 13–16); SS4, Roles and responsibilities (items 17–19) [20].

Each element was evaluated on a Likert scale from 1 to 5 where 5 = Strongly agree, 4 = Agree, 3 = Undecided, 2 = Disagree, 1 = Strongly disagree. A higher total score was associated with a greater student attitude/readiness to collaboratively learn with students from other professions [20].

### 2.3. Phase One: Translation and Cross-Cultural Adaptation of the RIPLS

The guidelines proposed by Beaton et al. [33] for cross-cultural adaptation were employed to develop the Italian version of the RIPLS questionnaire. Before starting the linguistic validation process, authorization was granted from the scale’s authors in 2020 [20]. After the authorization, the original version of the questionnaire was sent to two different official English/Italian translators (T1 and T2), with T1 being an expert in the topic and T2 without previous knowledge of the subject. The two translators did not know each other previously.

Afterwards, the respective translations were compared by six Italian healthcare interprofessional expert professors at different Albanian universities (where educational programs are delivered in Italian), thus leading to the drafting of a final version (version 3).

The new document was subsequently sent to an expert Italian professor in IPE, who, after a thorough check, found some wording inconsistencies in some items that were improved, as shown in Table 2.

After such modifications and a formal revision of version three, the final version in the Italian language was achieved and renamed version four.

This version was then subjected to a backward translation (version 5) in English to verify the compatibility between the version obtained and the original one. The work was conducted by a professor fluent in English and Italian at the University of Our Lady of Good Counsel (Tirane–Albania) (UNIZKM), who retranslated the questionnaire into the original language. The same six healthcare experts examined and compared the two versions sent to them: version 4 (Italian) and version 5 (English back-translated). The experts, after careful examination, recognized the equivalence of the two forms. A pre-test phase of the final translated version was carried out on 58 students to verify its intercultural comprehensiveness, resulting in a complete understanding of the instrument (Phase 5 pre-test).

The 5 phases for translating and adapting the scale are reported in Figure 1.

### 2.4. Phase Two: Data Collection Procedure

The cross-sectional collection of data was conducted among the eligible students (*n* = 1802) who were enrolled in courses in medicine, physiotherapy, nursing, and dentistry at “Our Lady of Good Counsel” (UNIZKM)-University of Tirane, Albania, April-June 2020. Since the UNIZKM has a partnership with the University “Tor Vergata”, Rome, Italy, and the official language of teaching is Italian, the data were collected in the Italian language. The sample size required a minimum of 10 participants per item for obtaining a suitable correlation or covariance matrix for performing factor analysis [34]; therefore, a minimum sample of 190 subjects was required to perform the EFA and other 190 to cross-validate the results derived from the EFA by employing a CFA. Overall, the study aimed to recruit at least 380 participants, where the first 190 responders were included in the sample for performing the EFA, and the second group of responders determined the sample for performing the CFA. As no strategies for randomizing the sample have been employed for selecting participants, the sample selection was based on a convenience sampling strategy, where all the eligible students were invited to participate (*n* = 1802), and the responders represented the final sample for the study. A google form via an internal mailing-controlled system at UNIZKM was sent to the eligible students, and the only criterion to determine the group for performing the EFA and the one for the CFA was the order in the responses of participants, acknowledging that the email with the google form was sent simultaneously to the entire population of eligible students.

### 2.5. Data Analysis

Before performing the EFA in the first group of responders, an item analysis via corrected item-total correlations was computed to make a preliminary selection of those items that better supported a stable assessment of the underlying measurement. Therefore, the analysis of corrected item-total correlations helped to reduce items whose elimination improved reliability coefficient alpha. Items that supported the highest reliability of the scale for determining the correlation matrix for the EFA were retained, also avoiding the risk of multicollinearity [35]. Although this preliminary analysis is not required to strictly create an adequate squared matrix for factor analyses, because there is an availability of robust estimation methods, items with poor item-total correlations could compromise the scale’s validity, consequently reducing its capacity to identify a stable construct in Italian-speaking settings [36]. Owing to this possibility, to reduce the items before the validity test [36,37], and acknowledging possible benefits derived from having a briefer scale with higher internal consistency, the items with an item-total correlation <0.30 or >0.90 were removed to enhance the reliability before the validity tests [37]. After removing uncorrelated items, the assumptions underpinning the possibility of performing an Exploratory Factor Analysis (EFA) were tested: systematic outliers were assessed using a visual inspection based on the observed univariate distribution and removed [38], linearity between items was assessed using bivariate scatterplots [39], and the Mardia’s test for multivariate normality was performed in Amos environment [40]. In addition, as each item’s measurement level was ordinal, a polychoric correlation matrix was used instead of the default Pearson’s matrix by employing the macro developed by Basto and Pereira [41]. EFA was conducted using the robust maximum likelihood estimator in the first 50% of the responder’s group. Bartlett’s test was used to assess the factorability of the correlation matrix and the Kaiser–Meyer–Olkin (KMO) index to assess the sample adequacy for factor analysis [42]. The eigenvalues according to their magnitude (scree plot) were first plotted to determine when the slope of the graph changed from steep to flat so that only the factors before the angle were kept as probable when determining the number of significant dimensions of the I-RIPLS. Additionally, the model derived from the scree plot interpretation was statistically assessed by extracting factors with eigenvalues >1. Parallel analysis with Monte Carlo simulations was utilised to confirm the most plausible factor structure hypothesized from eigenvalues, scree plot, and previous literature [43]. The parallel analysis was based on an adaptation of the SPSS syntax developed by O’Connor [44]. An oblique (Promax) rotation of the matrix, where items were Kaiser normalized, helped identify the associations between responses to the items (observed variables) and factors once the number of factors had been established. According to previous simulation and methodological studies [43,45], the number of factors was tested using Horn’s parallel analysis method. This method is performed using simulated random data to estimate the number of factors: along with the actual (real) data set; a random simulated (artificial) data set is created using the Monte Carlo Simulation Technique, and the estimated eigenvalues are determined. The number of factors where the eigenvalue in the simulated sample is greater than that of the actual data is regarded as significant when the approach is applied.

Subsequently, in the second 50% of the responders, a Confirmatory Factor Analysis (CFA) model using the maximum likelihood estimator was performed to cross-validate the factor structure derived from EFA. The χ^2^, the ratio between χ^2^ and degrees of freedom, the comparative fit index (CFI), the Tucker–Lewis index (TLI), the Root Mean Square Error of Approximation (RMSEA), and the Standardized Root Mean Square Residual (SRMR) have been employed as fit indexes to explain how well the confirmatory model explains the sample statistics. Adequate fit indexes were [46,47]: CFI ≥ 0.90, TLI ≥ 0.90, RMSEA lower than 0.080, and SRMR lower than 0.1. Additional specifications to the model have been evaluated using the modification index (Lagrange multiplier) to evaluate whether a specification would significantly affect the model’s ability to explain the sample statistics, following the procedure indicated by Whittaker [48]. When a single parameter restriction is removed from the model, the modification index is an estimate of how much the χ^2^ would be reduced, improving the model’s fit; a smaller χ^2^ shows that the model extensively explains the observed sample statistics. Reliability was assessed using the internal consistency of the I-RIPLS by computing Cronbach’s α for each subscale and the overall scale. As per the scoring procedure, each subscale score (factor) was calculated by summing the included items. Analytics were performed using IBM SPSS^®^ Statistics for Windows version 22 (IBM Corp., Armonk, NY, USA) and Mplus 8.1 (Muthén & Muthén, 1998–2017).

### 2.6. Ethical Consideration

The approval of the study protocol was in accordance with Italian and Albanian laws. Before using and starting our study, the authors of the original RIPLS instrument were contacted by email and granted the use of the RIPLS scale. The study was designed, conducted, registered, and reported consistently with the international ethical and scientific quality standards indicated by Good Clinical Practice (GCP) and Standard Operating Procedures (SOP). All participants were voluntarily involved and fully informed of the study’s purpose. They were asked to provide written informed consent. Participants were also informed of the confidentiality and anonymity of their responses during the data collection and analysis processes. This study was ethically approved by the Centre of Excellence for Nursing Scholarship OPI Rome protocol number 1.20.6.

## 3. Results

### 3.1. Descriptive Characteristics of the Sample

Valid questionnaires completed by students were 451 (25% of the entire population of students from the UNIZKM). Only respondents who completed the full questionnaire were considered valid for analyses, and 37 questionnaires were excluded because the analysis demonstrated that the missing values were higher than the 5% of the questionnaire. Overall, students who participated in this research were mostly female (*n* = 240, 58.0%) and mostly from the age class 19–27 (*n* = 358, 86.5%). The proportion of physiotherapy students (*n* = 219, 52.9%) was higher than students of other disciplines, nursing (*n* = 34, 8.2%), medicine (*n* = 120, 29%) and dentistry (*n* = 41, 9.9%).

The description of the characteristics of the first 207 responders, the group where the EFA has been performed, and those of the second 207 responders, the responses used for the CFA, are summarized in Table 3.

### 3.2. Preliminary Item Analysis

Five items were excluded in the item-total correlation analyses because the correlation score was weak <0.30 (items 10, 11, 12, 18 and 19). The corrected item-total correlations of the I-RIPLS are reported in Table 4.

### 3.3. Exploratory Factor Analysis

The Bartlett’s test of sphericity was significant (χ^2^_(91)_ = 2546.825; *p* < 0.001), and the Kaiser–Meyer–Olkin measure was 0.935, indicating that the sample was adequate for the factor analysis.

The most suitable solution from the EFA interpreting the scree plot and extracting factors with eigenvalues greater than 1.0 was a two-factor solution, which was confirmed by applying a parallel analysis and creating a Monte Carlo simulation following the procedure described by O’Connor [44]. In this model, all factor loadings were greater than 0.40 and accounted for a cumulative variance of 45.90%. Factor 1 was labelled as Teamwork Collaboration, based on the first version of the RIPLS [14] and the meaning of the items, which kept nine items (items 1, 5, 6, 7, 2, 8, 3, 9, 4) (explained variance of the rotated factor = 27.14%). Factor 2 was labelled considering the meaning of the kept items as Positive Professional Identity, Roles and Responsibility, and it included five items (items 15, 14, 16, 17, 13) which explained the variance of the rotated factor equal to 18.75%. The results of the factor analysis are shown in Table 5.

### 3.4. Confirmatory Factor Analysis

The unconstrained CFA showed that the two-factor structure of the I-RIPLS, to cross-validate the dimensionality derived from the EFA, produced an acceptable fit to the sample statistics (χ^2^_(76)_ = 172.719, *p* < 0.001; CFI = 0.903; TLI = 0.884; χ^2^/DF = 2.231; RMSEA = 0.078 [IC 90% = 0.063–0.094]; SRMR = 0.051; CFI = 0.927; TLI = 0.884). The factor loadings for each factor were all higher than 0.55 (Figure 2). However, by exploring possible specifications to the model, the residuals of item 4 and item 10 have been correlated by accounting for the modification index and the wording of the two items. The constrained model explained well the sample statistics (χ^2^_(75)_ = 154.672, *p* < 0.001; CFI = 0.920; TLI = 0.903; χ^2^/DF = 2.062; RMSEA = 0.072 [IC 90%= 0.056–0.088]; SRMR = 0.050) and Figure 2 shows the factor loadings.

### 3.5. Reliability of the Italian RIPLS

The alpha coefficient also estimated the internal consistency of the two dimensions regarding: “Teamwork and collaboration” (item 1–9), which was α = 0.876 and “Positive professional identity, roles and responsibility” (item 10–14), which was α = 0.833.

## 4. Discussion

The study aimed to develop the I-RIPLS, testing its cross-cultural linguistic validation and its validity and reliability to give educators the possibility to assess the student’s readiness to learn together with students from other professions in a collaborative way in the Italian educational contexts. The I-RIPLS is a self-report tool composed of 14 items with a 5-point scale, and the sum of the total scores ranges from 14 to 70. The adaptation of the RIPLS is consistent with the arguments of several authors stating that using a questionnaire in a country or in a specific language other than the one in which it was created necessarily requires that the literal translation is accompanied by intercultural adaptations [2,49]. For this reason, this study described the process of providing a linguistic and cultural adaptation by a group of experts.

Thus far, the RIPLS, a widely used scale that measures the readiness of health care students for shared learning via the original version developed by Parsell and Bligh [50], was redefined in its dimensionality by McFadyen [20]. Table 1 provides, in this regard, the synthesis of the several versions of the RIPLS over time by accounting for the number of final dimensions (subscales), participants in the psychometric testing (number), number of final items, internal consistency for the entire scale (Cronbach’s Alpha), and language availability; the reliability and validity results from our study are consistent with the previous studies. The presence of several versions with slightly different characteristics should be addressed with robust international studies aimed at providing cross-cultural multigroup invariance. In other words, future research might support the development of an updated version of the RIPLS that measures the same theoretical constructs.

The I-RIPLS has proven reliable and valid for 2 subscales, “Teamwork and Collaboration” and “Positive Professional Identity, Roles and Responsibility”, which showed good dimensionality. This factorial solution was very similar to the one of the original English version of the scale [25]. Furthermore, the two emerging subscales (domains) from the factorial analyses showed adequate internal consistency, and this aspect sustains the dimensionality of the I-RIPLS.

Thannauser et al. [17] have contended that the conceptualization of the language and constructs to be measured necessitates strong theoretical underpinnings instead of simply modifying or repattern existing tools. Some elements of roles and responsibilities require more detailed individual inspection due to their lack of conceptual coherence, as also this study demonstrates. For this reason, we suppose that a new reconceptualization of the RIPLS scale is needed in future years, even if this instrument showed good validity and reliability in the Italian setting. This further reconceptualization might facilitate the generalizability of the assessments performed with the updated scale versions.

This study has several limitations. First, data collection involved only one academic institution, which is located outside Italy despite featuring Italian as its official language. This aspect implies that although the students could fluently speak and read Italian, some were not native speakers, which might limit the generalizability of the results. Moreover, in this study, the enrollment year was not considered, and we administered the instrument to students of different learning years of the various health programmes. For example, we may expect students attending their third, fourth, or fifth year in medicine to have gathered different knowledge or collaboration skills during their past courses. Larger samples of clinical students from the same years would have allowed comparing results among more homogenous groups.

Finally, the impact of COVID-19 on IPE programs has caused a complex collection of data and may have biased some of the results. The pandemic has been changing how people live, making it more challenging to do interprofessional learning and work [51]. Another relevant aspect that might act as a limitation was the preliminary decision to remove from the scale the items showing non-significant corrected item-total correlations. Even if this approach is consistent with previous research [36,37], we have to acknowledge that the availability of robust methods for estimates of multivariate analyses would have been adequate to manage an initial validation assessment prior to performing the reliability assessments. An exploratory factor analysis would also have been adequate to address the possible deletion of ambiguous items. We preferred to perform a preliminary reliability assessment to optimize the subsequent psychometric evaluations of the items following the simulation study of Zijlmans and colleagues that supported the idea of performing a corrected item-total correlation to omit items from the preliminary tests before validity assessments [52]. The model specification means that the confirmatory approach became more exploratory than the unspecified model; therefore, future corroborations in different samples are needed for defining the psychometric performances of this new version of the I-RIPLS encompassing 14 items and two factors.

Additional limitations are related to the slightly different compositions of the sample used to perform the CFA because no medicine or dental students were present in that sample; this limitation implies that future tests have to consider a balanced sample of students from several healthcare courses and also measurement invariance tests are required. Furthermore, although the rigorous linguistic and cultural adaptation tends to mitigate diverse perceptions regarding the content of the items, in the current study, five items in the preliminary corrected item-total correlations were uncorrelated with the scale (without each item). This aspect implies that some cultural and/or linguistic aspects related to the excluded items may be investigated in-depth in future research to understand better if these items have to be excluded without compromising the collected information on the student’s readiness to learn with students from other professions.

## 5. Conclusions and Future Implications

The validated I-RIPLS might represent an efficient assessment of the student’s readiness to learn concurrently with students from other professions, showing evidence of validity and reliability. This research contributes to the intensification of focusing on IPE in Italian educational contexts. The results obtained show that the instrument is easy to comprehend and can be proposed as an interesting means of evaluating the attitudes of students from different courses to learning together. Research on the effectiveness of interprofessional education has consistently revealed that this educational approach is one of the best ways to prepare health professionals to cooperate in the different stages of professional life. Assessing their readiness is pivotal to facilitating educational strategies toward IPE. We consider that it would be appropriate to conduct future studies on roles and responsibilities with students of equivalent learning years in clinical settings. Moreover, a new reconceptualization of the instruments in different cultural settings is advised due to the variability of the dimensions that compose this tool in various contexts.

## Figures and Tables

**Figure 1 healthcare-10-01698-f001:**
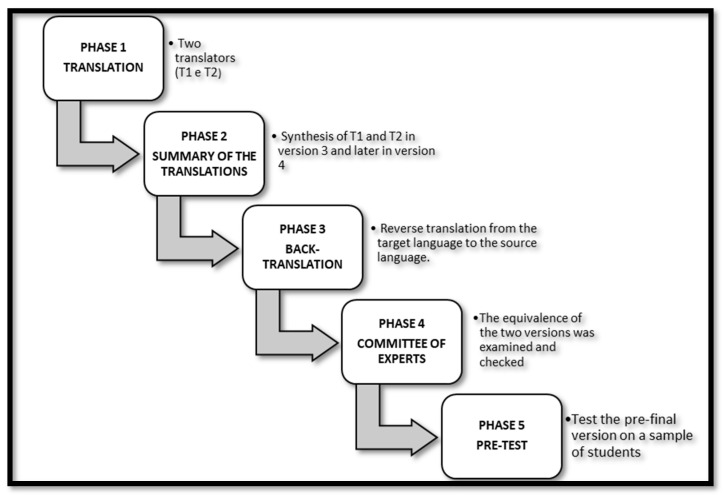
Guidelines for the transcultural adaptation of the RIPLS.

**Figure 2 healthcare-10-01698-f002:**
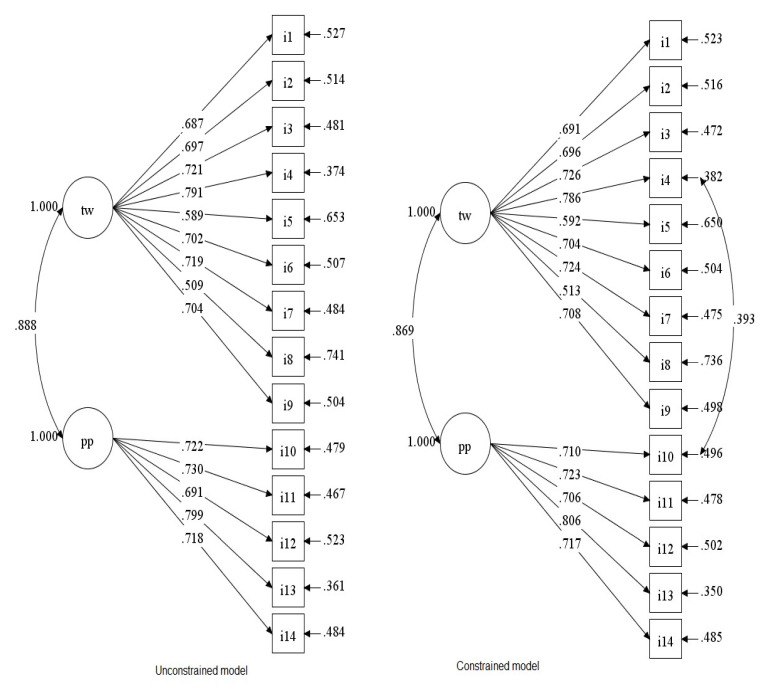
Unconstrained and constrained CFA models.

**Table 1 healthcare-10-01698-t001:** RIPLS characteristics and coefficient of reliability from previous studies.

Studies	Number of Final Subscales	Number of Participants	Number of Final Items	Cronbach’s Alpha for the Entire Scale	Language Availability
Parsell, Stewart, Bligh (1998) [25]	2	914	19	-	English
Parsell & Bligh (1999) [15]	3	120	19	0.90	English
McFadyen et al. (2005) ª [20]	4	308	19	0.84	English
Reid et al. (2006) ᵇ [16]	3	66	23	0.76	English
El-Zubeir et al. (2006) ᵇ [19]	3	178	20	0.61	Arabic
Lauffs et al. (2008) [23]	3	214	19	-	Swedish
Tamura et al. (2012) ᵇ [26]	3	132	19	0.74	Japanese
Tyastuti, D. et al. (2014) [27]	3	378	16	0.87	Indonesian
Peduzzi, M. et al. (2015) [18]	3	327	27	-	Portuguese/Brazil
Cloutier, J. et al. (2015) [21]	3	141	16	0.90	French
Mahler, C. et al. (2016) [22]	3	531	19	-	German
Pype, P. et al. (2016) [28]	3	510	23	0.88	Dutch
Nørgaard, B. et al. (2016) [29]	4	570	29	-	Danish
Oishi, A. et al. (2017) [1]	4	368	23	0.7	Japanese
Ergonul, E. et al. (2018) [24]	3	213	19	0.85	Turkish
Li, Z. et al. (2018) [11]	4	282	19	0.70	Chinese
Milutinović, D. et al. (2018) [12]	2	257	19	0.90	Serbian
Ataollahi, M. et al. (2019) [30]	4	200	19	0.94	Persian
Torsvik et al. (2021) [31]	4	307	19	0.85	Norwegian
Villagrán, I et al. (2022) [32]	3	407	24	0.86	Spanish

ᵃ McFadyen’s studies produced a four-factor model. ᵇ All of these studies produced three factors, but the elements that made up these factors were different from other studies.

**Table 2 healthcare-10-01698-t002:** Linguistic and cultural adaptation of the RIPLS (final Italian version and last modifications in italics).

Original Items	*Translated Items*
1. Learning with other students and professionals will make me a more effective health and social care team member.	*1. Imparare con altri studenti/professionisti mi renderà un membro più efficace di una squadra di assistenza sanitaria e sociale.*
2. Patients would ultimately benefit if health and social care student professionals worked together.	*2. Alla fine i pazienti trarrebbero beneficio se gli studenti e i professionisti dell’assistenza sanitaria e sociale lavorassero insieme.*
3. Shared learning with other health and social care students professionals will increase the ability to understand clinical problems.	*3. L’apprendimento condiviso con altri studenti/professionisti della sanità e dell’assistenza sociale aumenterà le mie capacità di comprendere i problemi clinici.*
4. Communications skills should be learned with other health and social care students/professionals.	*4. Le capacità di comunicazione dovrebbero essere apprese con altri operatori sanitari e sociali.*
5. Teamworking skills are vital for all health and social care students/professionals to learn.	*5. Le capacità di lavorare in gruppo sono vitali per tutti gli studenti/professionisti dell’assistenza sanitaria e sociale per apprendere.*
6. Shared learning will help to understand my own professional limitations.	*6. L’apprendimento condiviso mi aiuterà a comprendere i miei limiti professionali.*
7. Learning between health and social care students before qualification and for professionals after qualification would improve working relationships after qualification/collaborative practice.	*7. L’apprendimento tra gli studenti dell’assistenza sanitaria e sociale, prima e dopo la qualifica professionale, migliorerebbe i rapporti di lavoro e la pratica collaborativa.*
8. Shared learning will help me think positively about other health and social care professionals.	*8. L’apprendimento condiviso mi aiuterà a pensare positivamente verso gli altri professionisti sanitari e sociali.*
9. For small group learning to work, students/professionals need to respect and trust each other.	*9. Per imparare a lavorare in piccoli gruppi, studenti e professionisti devono rispettarsi e fidarsi l’uno dell’altro.*
10. I do not want to waste time learning with other health and social care students/professionals.	*10. Non voglio perdere tempo a imparare con altri studenti e professionisti sanitari e sociali.*
11. It is not necessary for undergraduate/postgraduate health and social care students/professionals to learn together.	*11. Non è necessario che gli studenti laureati e post laureati nell’assistenza sanitaria e sociale apprendano insieme.*
12. Clinical problem solving can only be learnt effectively with students/professionals from my own school/organization.	*12. La risoluzione dei problemi clinici può essere appresa efficacemente solo con studenti/professionisti della mia scuola/organizzazione.*
13. Shared learning with other health and social care professionals will help me to communicate better with patients and other professionals.	*13. L’apprendimento condiviso con altri professionisti sanitari e sociali mi aiuterà a comunicare meglio con i pazienti e gli altri professionisti.*
14. I would welcome the opportunity to work on small group projects with other health and social care students/professionals.	*14. Gradirei l’opportunità di lavorare su progetti in piccoli gruppi con altri studenti di assistenza sanitaria e sociale.*
15. I would welcome the opportunity to share some generic lectures, tutorials or workshops with other health and social care students/professionals.	*15. Gradirei l’opportunità di condividere alcune lezioni di base, tutorial o seminari con altri studenti/professionisti dell’assistenza sanitaria e sociale.*
16. Shared learning and practice will help me clarify the nature of patients’ or clients’ problems.	*16. L’apprendimento e la pratica condivisi mi aiuteranno a chiarire la natura dei problemi dei pazienti.*
17. Shared learning before and after qualification will help me become a better team worker.	*17. L’apprendimento condiviso prima e dopo la qualifica professionale mi aiuterà a diventare un miglior collaboratore.*
18. I am unsure what my professional role will be/is.	*18. Non sono sicuro di quale sia/sarà il mio ruolo di professionista.*
19. I have to acquire much more knowledge and skill than other students/professionals in my own faculty/organization.	*19. Devo acquisire molte più conoscenze e abilità rispetto ad altri studenti/professionisti nella mia facoltà/organizzazione.*

**Table 3 healthcare-10-01698-t003:** Characteristics of the responders (*n* = 414).

	Tot (*n* = 414)		EFA (*n* = 207)		CFA (*n* = 207)	
	*n*	%	*n*	%	*n*	%
**Disciplines**						
Physiotherapy	219	52.9	29	14.0	190	91.8
Nursing	34	8.2	17	8.2	17	8.2
Medicine	120	29.0	120	58.0	-	-
Dentistry	41	9.9	41	19.8	-	-
**Sex**						
Females	240	58.0	147	71.0	93	44.9
Males	174	42.0	60	29.0	114	55.1
**Age**						
Years (Mean; standard deviation)	24.37	4.23	25.1	4.06	23.64	4.28

**Table 4 healthcare-10-01698-t004:** Corrected item-total correlation and alpha if items deleted (RIPLS).

	1° Item Analysis		2° Item Analysis	
	Corrected Item-Total Correlation	Cronbach’s Alpha if Item Deleted	Corrected Item-Total Correlation	Cronbach’s Alpha if Item Deleted
**RIPS1**	**0.578**	**0.798**	**0.602**	**0.907**
**RIPS2**	**0.528**	**0.799**	**0.586**	**0.907**
**RIPS3**	**0.580**	**0.797**	**0.604**	**0.907**
**RIPS4**	**0.536**	**0.798**	**0.576**	**0.908**
**RIPS5**	**0.608**	**0.794**	**0.627**	**0.906**
**RIPS6**	**0.570**	**0.795**	**0.608**	**0.907**
**RIPS7**	**0.691**	**0.790**	**0.731**	**0.902**
**RIPS8**	**0.612**	**0.792**	**0.691**	**0.903**
**RIPS9**	**0.538**	**0.799**	**0.549**	**0.909**
RIPS10	0.149	0.824		
RIPS11	0.197	0.820		
RIPS12	0.021	0.835		
**RIPS13**	**0.618**	**0.793**	**0.641**	**0.905**
**RIPS14**	**0.545**	**0.797**	**0.602**	**0.907**
**RIPS15**	**0.561**	**0.796**	**0.587**	**0.907**
**RIPS16**	**0.570**	**0.797**	**0.608**	**0.906**
**RIPS17**	**0.680**	**0.792**	**0.712**	**0.903**
RIPS18	−0.130	0.843		
RIPS19	0.164	0.820		

**Table 5 healthcare-10-01698-t005:** I-RIPLS Exploratory Factor Analysis.

	New Item Scale	Former Item Scale	Factor 1	Factor 2	Communalities	Cronbach’s Alpha
Factor 1-Teamwork & collaboration	I-RIPLS 1	RIPLS1	**0.722**	−0.089	0.584	0.883
	I-RIPLS 2	RIPLS5	**0.756**	−0.087	0.537	
	I-RIPLS 3	RIPLS6	**0.614**	0.002	0.519	
	I-RIPLS 4	RIPLS7	**0.782**	0.016	0.622	
	I-RIPLS 5	RIPLS2	**0.462**	0.243	0.465	
	I-RIPLS 6	RIPLS8	**0.67**	0.117	0.565	
	I-RIPLS 7	RIPLS3	**0.415**	0.195	0.476	
	I-RIPLS 8	RIPLS9	**0.623**	0.040	0.405	
	I-RIPLS 9	RIPLS4	**0.331**	0.236	0.425	
Factor 2-Positive Professional Identity, Roles and Responsibility	I-RIPLS 10	RIPLS15	−0.008	**0.661**	0.654	0.818
	I-RIPLS 11	RIPLS14	−0.108	**0.773**	0.632	
	I-RIPLS 12	RIPLS16	−0.032	**0.760**	0.608	
	I-RIPLS 13	RIPLS17	0.182	**0.621**	0.645	
	I-RIPLS 14	RIPLS13	0.270	**0.418**	0.515	

## Data Availability

The data presented in this study are available within the article.

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
