# Peer review of "Italian Translation and Validation of the Readiness for Interprofessional Learning Scale (RIPLS) in an Undergraduate Healthcare Student Context"

_healthcare, 2022, doi:10.3390/healthcare10091698_

Round 1
Reviewer 1 Report
Overall, a nice paper about translating RIPLS into Italian.
The introduction is well written. Just a few questions about the references for this section. For Line 50 - reference 2 is appropriate. Reference number 1 is not needed - and maybe there was supposed to be a different reference here since this is a duplicate reference to number 22?
Line 60 - recommend changing to a different reference that better supports this statement. Ref 8 is about validity of the RIPLS in Turkish students not an increase in patient care quality or efficiency.
Line 62 - recommend changing to a different reference that better supports this statement. Reference 10 is about converting the RIPLS to Chinese.
Methods - nice explanation of what was done and references to back up method choices.
Line 131 seems to stop short - there were inconsistencies in some items and directs us to Table 1. But it is not clear what were the inconsistencies.
Line 146: recommend to add of translation after the 5 phases (since there are phases of translation and in the study methods - This helps to clarify).
Results:
Table 2 - Is there any concern that there were no Medicine or Dental students in the CFA? Should that be addressed in the discussion?
Table 3 - can you please add to the discussion any concerns or thoughts about why all the negative professional items had to be excluded? Is this a cultural result or could it be due to the translation?
Line 251 - should that be a period for the p-value and not a comma?
Figure 2 - could maybe delete this.
Table 6 - recommend moving this out of the results. It should be in the discussion section. It is a nice review of the current data. Or if keep it in the results, then add something about this to the methods.
LIne 277 - recommend to remove the before IPE
Line 278 - add s to end of 14-item
Line 279 remove item from total item scores range
Line 280 - is that statement correct - does the tool measure reliability and applicability?
After line 284, recommend discussing Table 6 - how does your adaptation compare to other studies reliability scores?
Lines 312 -313 - could this also be a strength - since you have more ability to generalize?
Line 320 change difficult the interprofessional to difficult to do the interprofessional..
Thanks for the opportunity to review your article.
Author Response
Authors. Thank you immensely for the opportunity to continue to develop our paper. We appreciate the valuable comments and suggestions provided by you, and we have considered all the points and have revised our manuscript according to them. We will be happy to provide more information and clarification concerning our manuscript if needed.
Overall, a nice paper about translating RIPLS into Italian.
Authors: Thank you for having appreciated our study.
The introduction is well written. Just a few questions about the references for this section. For Line 50 - reference 2 is appropriate. Reference number 1 is not needed - and maybe there was supposed to be a different reference here since this is a duplicate reference to number 22?
Authors: Thank you for having pinpointed these inaccuracies. We have updated the references and the typos for references.
Line 60 - recommend changing to a different reference that better supports this statement. Ref 8 is about validity of the RIPLS in Turkish students not an increase in patient care quality or efficiency.
Line 62 - recommend changing to a different reference that better supports this statement. Reference 10 is about converting the RIPLS to Chinese.
Authors: Thank you for having identified these inaccuracies. We changed the references to better support the statements.
Methods - nice explanation of what was done and references to back up method choices.
Authors: Thank you for having appreciated our methods.
Line 131 seems to stop short - there were inconsistencies in some items and directs us to Table 1. But it is not clear what were the inconsistencies.
Authors: We have clarified the meaning as follows: The new document was subsequently sent to an expert Italian professor in IPE, who, after a thorough check, found some wording inconsistencies in some items that were improved, as shown in Table 2.
Line 146: recommend adding of translation after the 5 phases (since there are phases of translation and in the study methods - This helps to clarify).
Authors: Thank you for this comment; we clarified that the 5 phases for translating and adapting the scale are reported in Figure 1.
Results:
Table 2 - Is there any concern that there were no Medicine or Dental students in the CFA? Should that be addressed in the discussion?
Authors: thank you for this great point. We added a discussion of this limitation in the current version of the manuscript.
“Additional limitations are related to the slightly different compositions of the sample used to perform the CFA because no medicine or dental students were present in that sample; this limitation implies that future tests have to consider a balanced sample of students from several healthcare courses, and also measurement invariance tests are required.”
Table 3 - Can you please add to the discussion any concerns or thoughts about why all the negative professional items had to be excluded? Is this a cultural result or could it be due to the translation?
Authors: Thank you for this comment. We are not able to exclude both possibilities, and for this reason, we added both options to the discussion.
“[…]Furthermore, although the rigorous linguistic and cultural adaptation tends to mitigate diverse perceptions regarding the content of the items, in the current study, five items in the preliminary corrected item-total correlations were uncorrelated with the scale (without each item). This aspect implies that some cultural and/or linguistic aspects related to the excluded items may be investigated in-depth in future research to understand better if these items have to be excluded without compromising the collected information on the student's readiness to learn with students from other professions.
Line 251 - Should that be a period for the p-value and not a comma?
Authors: Corrected.
Figure 2 - Could maybe delete this.
Authors: We prefer to keep this to show the factor loadings in the specified and unspecified models.
Table 6 - Recommend moving this out of the results. It should be in the discussion section. It is a nice review of the current data. Or if keep it in the results, then add something about this to the methods.
Authors: We moved it as a background, as also suggested by reviewer#3.
Line 277 - Recommend removing the before IPE
Authors: we change IPE with “students’ readiness to learn together with students from other professions”.
Line 278 - Add s to end of 14-item
Authors: Corrected.
Line 279 Remove item from total item scores range
Authors: Corrected.
Line 280 - Is that statement correct - does the tool measure reliability and applicability?
Authors: Thank you; we edited the statement.
After line 284, recommend discussing Table 6 - how does your adaptation compare to other studies reliability scores?
Authors: We added a brief discussion with the table that in the current version is in the background.
Lines 312 -313 - could this also be a strength - since you have more ability to generalize?
Authors: We think so; in the current version, we have better highlighted this aspect.
Line 320 - Change difficult the interprofessional to difficult to do the interprofessional.
Authors: Corrected.
Thanks for the opportunity to review your article.
Authors: Thank you for your great input.

Reviewer 2 Report
Well written paper.
Interesting exercise regarding the use of tool in Italian contexts reported very well using appropriate and correct language throughout.
Nonetheless “imagine” on Ln307 may be changed to “ suppose”.
The conclusion refers and assumes to the impact of the use of the tool to evaluate inter professional learning . This is not so much the case since the tool seems to evaluate the readiness for interprofessional learning . Hence the inclusion may need some tweaking. Reference to “this technique” Needs to be rephrased because “technique” falls short of capturing and representing the concept of IPE, and “ means of evaluating interprofessional education “ needs to be removed because it misleads the reader in that this text suggests that the tool seems to evaluate IPE not readiness for IOE whereby readiness is believed to be a requisite for effective IOE.
Author Response
Thank you immensely for the opportunity to continue to develop our paper. We appreciate the valuable comments and suggestions provided, and we have considered all the points and have revised our manuscript according to them. We will be happy to provide more information and clarification concerning our manuscript, if needed.
Well written paper.
Interesting exercise regarding the use of tool in Italian contexts reported very well using appropriate and correct language throughout.
Authors: Thank you for having appreciated our study.
Nonetheless “imagine” on Line 307 may be changed to “suppose”.
Authors: Corrected.
The conclusion refers and assumes to the impact of the use of the tool to evaluate inter professional learning. This is not so much the case since the tool seems to evaluate the readiness for interprofessional learning. Hence the inclusion may need some tweaking. Reference to “this technique” Needs to be rephrased because “technique” falls short of capturing and representing the concept of IPE, and “means of evaluating interprofessional education “needs to be removed because it misleads the reader in that this text suggests that the tool seems to evaluate IPE not readiness for IOE whereby readiness is believed to be a requisite for effective IOE.
Authors: Thank you for this great comment. We fully agree and updated the current version.

Reviewer 3 Report
In carefully evaluating the manuscript, I made a number of comments and suggestions that I would like to share with you to contribute to the study. These are suggestions to improve the work, especially with regard to psychometric rigor in data analysis. I hope they're helpful.
ABSTRACT
Lines 32-34: “The preliminary 32 inter-item and item-total correlations showed that five items were not correlated and were removed 33 from the correlation matrix before testing the dimensionality of the I-RIPLS with factor analysis.”
This information seems strange to me. First, because in this type of analysis, the correlation matrix is a square matrix, so one would expect 361 correlations between the items. Second, because I am not aware, in the robust and current psychometric literature, of recommendations for this type of procedure before carrying out the factor analysis. I comment further on this issue in the method section.
Lines 38-39: “The most plausible dimensionality from EFA, by acknowledging the interpretation of the scree plot, the 38 eigenvalues greater than 1.”
The K1 / scree plot criterion used to choose the number of factors to be retained has been heavily criticized in the psychometric literature for being subjective and ambiguous. I comment further on this issue in the method section.
INTRODUCTION
Lines 89-90: “The literature has reported that McFadyen’s version was translated into Brazilian/Portuguese”
The version validated in Brazil (Peduzzi et al., 2015) in the indicated study was not the 19-item version proposed by McHadyen’s et al (2005), but the 29-item version proposed by Reid et al. (2006) and validated, for example, by El-Zubeir et al. (2006) and Mattick and Bligh (2009). Consult better:
Peduzzi M., Norman, I.J., Coster, S., & Meireles, E. (2015). Cross-cultural adaptation and validation of the Readiness for Interprofessional Learning Scale in Brazil. Rev Esc Enferm USP, 49(Esp2), 7-15. https://doi.org/10.1590/S0080-623420150000800002
Reid, R., Bruce, D., Allstaff, K., & McLernon, D. (2006). Validating the Readiness for Interprofessional Learning Scale (RIPLS) in the postgraduate context: are health care professionals ready for IPL? Medical Education, 40(5), 415-422. https://doi.org/10.1111/j.1365-2929.2006.02442.x
El-Zubeir, M., Rizk, D.E.E., & Al-Khalil, R.K. (2006). Are senior UAE medical and nursing students ready for interprofessional learning? Validating the RIPL scale in a Middle Eastern context. Journal of Interprofessional Care, 20(6), 619-632. https://doi.org/10.1080/13561820600895952
MATERIALS AND METHODS
Line 95: materials and Methods – correct Materials
Lines 116-117: “A higher total score was associated with greater inter-professional learning”
RIPLS does not assess interprofessional learning per se, but the student's attitude/readiness to learn together with students from other professions in a collaborative way.
Line 151: “The cross-sectional collection of data was conducted among the eligible students (1802) who were enrolled in courses in medicine…”
Correct: (n = 1.802)
Data analysis (Lines 170-175)
The literature indicated (Rozental et al., 2016) does not seem to be sufficient and adequate to justify this procedure. I am not aware of technical works on psychometrics that guide this type of data treatment even before submitting the items to exploratory factor analysis – these multivariate analyses. In my assessment it makes no sense to make decisions about excluding items before assessing their multivariate association pattern in an exploratory factor analysis (EFA).
What is indicated is to verify a series of assumptions that the EFA requires, such as: presence and systematicity of outliers (there is no report in the text that was evaluated); multivariate normality (there is no report in the text that was evaluated); choice of estimators suitable for the measurement level (assumption that the study ignores when using Pearson matrices and more suitable estimators for interval data, when the RIPLS measurement level is ordinal, for which polychoric correlation matrices are the most indicated in analyzes factorials), multicollinearity (assuming that the study evaluates in a simplistic way and, in my opinion, wrong when it comes to preparing the database for an EFA); linearity (there is no report in the text that was evaluated).
Having verified these assumptions, the assessment of the retention or not of items must be based on the parameters estimated by the EFA, for example, the saturation of the item in the factor; the item-total correlation of the items in their respective factors and, obviously, the theoretical-interpretative plausibility.
The authors make this decision to exclude items based on bivariate analyzes and not on multivariate models capable of evaluating how well these items are grouped together with the others – element evaluated after the EFA, not before; assume unidimensionality of the set of items when calculating the item-total correlation.
Another problem that I identify in the methodological choices concerns the use of the K1 / scree plot criterion to choose the number of factors to be retained. This criterion has been heavily criticized in the psychometric literature for being subjective and ambiguous (see list below of works that can help). One study of psychometric simulation, in particular, demonstrated that the parallel analysis criterion (Horn, 1965) is one of the most accurate, indicating the correct number of factors in the matrix in 92% of the cases involved in the simulations – with the scree test, only 52% accuracy (Zwick & Velicer, 1986).
Anyway, in my opinion, this decision by the authors is wrong and compromises the entire list of results presented in the sequence.
Asún, R. A., Rdz-Navarro, K., & Alvarado,J. M. (2016). Developing multidimensional Likert scales using item factor analysis: The case of four-point items. Sociological Methods & Research, 45(1), 109-133. https://doi.org/10.1177/0049124114566716
Fabrigar, L. R., Wegener, D. T., MacCallum, R. C. & Strahan, E. J. (1999). Evaluating the use of exploratory factor analysis in psychological research. Psychological Methods, 4(3),272-299.
Hair, J.F.; Anderson, R.E.; Tatham, R.L.; Black, W.C. (2005). Multivariate Data Analysis, 6th ed., Prentice Hall, Englewood Cliffs, NJ
Hayton, J. C., Allen, D. G. & Scarpello, V. (2004). Factor retention decisions in exploratory factor analysis: A tutorial on parallel analysis. Organizational Research Methods, 7(2), 191-207.
Holgado–Tello, F. P., Chacón–Moscoso, S., Barbero–García, I., & Vila–Abad, E. (2010). Polychoric versus Pearson correlations in exploratory and confirmatory factor analysis of ordinal variables. Quality & Quantity, 44(1), 153.
Horn, J. L. (1965). A rationale and technique for estimating the number of factors in factor analysis. Psychometrika, 30(2), 179-185. https://doi.org/10.1007/BF02289447
Kline, R. B. (2011). Principles and practice of Structural Equation Modeling. 3th. The Guilford Press: New York, London.
Tabachnick, B. G., & Fidell, L. S. (2013). Using multivariate statistics (6th ed.) New Jersey: Pearson Education.
Zwick, W. R. & Velicer, W. F. (1986). Comparison of Five Rules of Determining the Number of Componentes to Retain. Multivariate Behavioral Research, 17, 258-269.
Line 194: “Adequate fit indexes were: CFI ≥ 0.90, TLI ≥ 0.90, RMSEA lower than 0.80, and SRMR lower than 0.1.”
It remains to refer to the literature that supports these criteria. In Kline (2011), for example, values are accepted for RMSEA < 0.08 and for SRMR < 0.06.
RESULTS
Table 3: It makes no sense to assume, before the dimensionality analysis / factor analysis, that the items are one-dimensional (to calculate Cronbach's Alpha this assumption must have been verified before!). If there is already robust evidence of this assumption of one-dimensionality of these items in the respective factors in the literature, why not start with a confirmatory approach? Exploratory analysis is used precisely to evaluate these groupings, excluding items before seems to me to be wrong, as explained above. Furthermore, a total Cronbach's alpha makes no sense because the scale does not have an overall score – this total score has never been modeled in the studies I know of with RIPLS, in any of its versions. There are also problems in assuming the Likert scale as a measure at an interval level, when in fact this scale measures at an ordered ordinal/categorical level.
Lines 238-239: “The most suitable solution from the EFA interpreting the scree plot and extracting factors with eigenvalues greater than 1.0 was a two-factor solution.”
As already mentioned, the use of this criterion is widely criticized in the psychometric literature. As the empirical eigenvalues obtained in the study were not presented, this reviewer is unable to perform the parallel analysis and better evaluate the results of the study. It is suggested that the eigenvalues obtained from the data matrix be presented.
Figure 2 and Lines 253-255: “However, by exploring possible specifications to the model, the residuals of item 4 and item 10 have been correlated by accounting for the modification index and the wording of the two items.”
By correlating errors, the authors again start with an exploratory approach. They misrepresent the true nature of AFC, falling back into an exploratory approach. This is because, by including correlation arrows between the errors, the chi-square value is reduced and all the model's adequacy indices are improved, so nothing is being “tested”. By including arrows “suggested” by the modification indices, confirmatory analysis is left aside, returning to an exploratory approach. For each arrow there should be a theoretical justification for its and, later, a new analysis with a validation sample should be performed. I suggest consulting:
Breckler, S. J. (1990). Applications of Covariance Structure Modeling in Psychology: Cause for Concern? Psychological Bulletin,107(2), 260-273.
Table 5: There are inaccuracies. The item-total correlation should only be calculated between the items of each factor, not for the scale as a whole. Furthermore, as already explained, it makes no sense to calculate a Cronbach's alpha for the entire scale - this factor has not been modeled. It is not possible to calculate the internal consistency or even calculate the score without unequivocal evidence of the unidimensionality of the set of items under analysis.
Table 6: In my opinion, the data presented in Table 6 are not the results of the analysis of data derived from the field research, but rather the literature review. Therefore, this Table does not seem to make sense here in the Results section – it would be better if it were presented in the Introduction of the manuscript.
Furthermore, I note that among the 20 studies reported, regardless of the versions/types of items, most propose three (n = 12) or four factors (n = 6) of readiness for IPE - only two studies proposed two factors to represent the structure of RIPLS. This fact, together with all the previous comments about methodological choices in psychometric analyses, lead me to more consistent assumptions about methodological inadequacies in the evaluation of the internal structure / dimensionality of the RIPLS presented in the manuscript.
Considering everything previously, my suggestion to redo the psychometric evaluations of using the most appropriate manuscript, according to the literature indicated throughout the evaluation and return for evaluation. Unless better evaluated, this is my opinion.
Author Response
Thank you immensely for the opportunity to continue to develop our paper. We appreciate the valuable and detailed comments and suggestions provided, and we have considered all the points and revised our manuscript according to them. We will be happy to provide more information and clarification concerning our manuscript if needed.
Reviewer: In carefully evaluating the manuscript, I made a number of comments and suggestions that I would like to share with you to contribute to the study. These are suggestions to improve the work, especially with regard to psychometric rigor in data analysis. I hope they're helpful.
Authors: Dear Reviewer, your review was awesome and actually very useful for improving our work.
ABSTRACT
Lines 32-34: “The preliminary 32 inter-item and item-total correlations showed that five items were not correlated and were removed 33 from the correlation matrix before testing the dimensionality of the I-RIPLS with factor analysis.”
This information seems strange to me. First, because in this type of analysis, the correlation matrix is a square matrix, so one would expect 361 correlations between the items. Second, because I am not aware, in the robust and current psychometric literature, of recommendations for this type of procedure before carrying out the factor analysis. I comment further on this issue in the method section.
Authors: Thank you immensely for this comment. Actually, the explanation of our approach in the previous version was inaccurate as some misinterpretations might arise from reading the phrases on the corrected item-total correlations (it was not an item-item correlation analysis).
We agree that the point in keeping this strategy was not the “suitability” of the matrix because with every available robust estimation method is possible to perform factor analyses as it is frequently performed by several researchers. In this particular situation, the possibility of performing a preliminary analysis on the item-to-total correlations was discussed in-depth between authors, and it was not so immediate to unravel. We opted for this preliminary analysis to delete items having poor item-total correlations that could compromise the scale’s validity, consequently reducing its capacity to identify a stable construct in Italian-speaking settings.
We edited the abstract as follows:
“Even their deletion was not mandatory for generating a suitable correlation matrix for factor analysis, the advantages of keeping only items contributing to a more stable measurement with a shorter scale represented the rationale for removing items with non-significant item-to-total correlation from the correlation matrix before testing the dimensionality of the I-RIPLS with factor analysis”.
Precisely, we edited the text (in the manuscript) as follows:
“Before performing the EFA in the first group of responders, an item analysis via corrected item-total correlations was computed to preliminary select those items that better supported a stable assessment of the underlying measurement. Therefore, the analysis of corrected item-total correlations helped to reduce items whose elimination improved reliability coefficient alpha to the ones that supported the highest reliability of the scale for determining the correlation matrix for the EFA, also avoiding the risk of multicollinearity [26]. Although this preliminary analysis is not required to strictly create an adequate squared matrix for factor analyses because there is an availability of robust estimation methods, items with poor item-total correlations could compromise the scale’s validity, consequently reducing its capacity to identify a stable construct in the Italian-speaking settings [27]. Owing to this possibility of reducing the items before the validity test, as per previous studies [27,28] and acknowledging possible benefits derived from having a briefer scale with higher internal consistency, the items with an item-total correlation < .30 or > .90 were removed for their effects on the reliability before the validity tests [28].”
In the limitation paragraph, we further discussed the cons at the net of the pros (having a shorter scale and likely more stable measure).
Here is some literature consistent with our approach, acknowledging that we recognize that most of the psychometric literature tests the validity without preliminary reliability assessments.
- Hinkin TR. A Review of Scale Development Practices in the Study of Organizations. Journal of Management. 1995;21(5):967-988. doi:10.1177/014920639502100509 [on page 975]
- Parasuraman, A., Zeithaml, V. A., Malhotra, A. (2005). ES-QUAL: A multiple-item scale for assessing electronic service quality. Journal of service Research, 7(3), 213-233. [on page 219]
- Barrera, R. B., García, A. N., Moreno, M. R. (2014). Evaluation of the e-service quality in service encounters with incidents: Differences according to the socio-demographic profile of the online consumer. Revista Europea de Dirección y Economía de la Empresa, 23(4), 184-193.
- Cristobal, E., Flavian, C., Guinaliu, M. (2007). Perceived e‐service quality (PeSQ): Measurement validation and effects on consumer satisfaction and web site loyalty. Managing Service Quality: An International Journal 17 No. 3, pp. 317-340. https://doi.org/10.1108/09604520710744326
Lines 38-39: “The most plausible dimensionality from EFA, by acknowledging the interpretation of the scree plot, the 38 eigenvalues greater than 1.”
The K1 / scree plot criterion used to choose the number of factors to be retained has been heavily criticized in the psychometric literature for being subjective and ambiguous. I comment further on this issue in the method section.
Authors: Thank you for this comment. We agree that only scree plot or eigenvalues (Kaiser-Guttmanrule method) could be criticized, even if we also considered a theoretical structure from the previous validation studies. To address your comment, we added Horn's parallel analysis to confirm the factor structure (as per your suggestion in subsequent comments).
Although there are several aspects to take into account when performing exploratory factor analysis, we are aware that this step is the most crucial and important one (variance ratios explained by factors, factor loadings of items, items with high factor loadings more than one factor, and so on). The worth of exploratory factor analysis depends on the ability to separate latent variables (factors) from the observed ones; therefore, choosing the number of factors is considerably more crucial than other considerations, such as choosing the analytical method or kind of rotation. We agree that determining the precise balance between correlations is absolutely crucial.
For these reasons, we added another strategy to identify the number of elements is "parallel analysis," which Horn (1965) proposed. The number of factors may be accurately determined using the parallel analysis method, according to studies accomplished in the 1970s. Simulated random data is used in parallel analysis to estimate the number of factors. Using a Monte Carlo Simulation Technique, a random simulative (artificial) data set is generated besides the actual (real) data set, and the estimated eigenvalues are calculated. When the method is employed, the number of factors where the eigenvalue in the simulative sample is higher than that of the actual data is considered significant.
Example of literature:
- Silverstein, A. B. (1977). Comparison of two criteria for determining the number of factors. Psychological Reports, 41, 387–390.
- Silverstein, A. B. (1987). Note on the parallel analysis criterion for determining the number of common factor or principal components. Psychological Reports, 61, 351–354.
INTRODUCTION
Lines 89-90: “The literature has reported that McFadyen’s version was translated into Brazilian/Portuguese”
The version validated in Brazil (Peduzzi et al., 2015) in the indicated study was not the 19-item version proposed by McFadyen’s et al (2005), but the 29-item version proposed by Reid et al. (2006) and validated, for example, by El-Zubeir et al. (2006) and Mattick and Bligh (2009). Consult better:
Peduzzi M., Norman, I.J., Coster, S., & Meireles, E. (2015). Cross-cultural adaptation and validation of the Readiness for Interprofessional Learning Scale in Brazil. Rev Esc Enferm USP, 49(Esp2), 7-15. https://doi.org/10.1590/S0080-623420150000800002
Reid, R., Bruce, D., Allstaff, K., & McLernon, D. (2006). Validating the Readiness for Interprofessional Learning Scale (RIPLS) in the postgraduate context: are health care professionals ready for IPL? Medical Education, 40(5), 415-422. https://doi.org/10.1111/j.1365-2929.2006.02442.x
El-Zubeir, M., Rizk, D.E.E., & Al-Khalil, R.K. (2006). Are senior UAE medical and nursing students ready for interprofessional learning? Validating the RIPL scale in a Middle Eastern context. Journal of Interprofessional Care, 20(6), 619-632. https://doi.org/10.1080/13561820600895952
Authors: Thank you for having noticed this aspect. We have amended the introduction accordingly.
MATERIALS AND METHODS
Line 95: materials and Methods – correct Materials
Authors: Thank you for having noticed this typo. We have emended it accordingly.
Lines 116-117: “A higher total score was associated with greater inter-professional learning”
RIPLS does not assess interprofessional learning per se, but the student's attitude/readiness to learn together with students from other professions in a collaborative way.
Authors: Thank you for this comment. We have amended the sentence accordingly.
Line 151: “The cross-sectional collection of data was conducted among the eligible students (1802) who were enrolled in courses in medicine…”
Correct: (n = 1.802)
Authors: Thank you for having noticed this typo. We have amended the point accordingly.
Data analysis (Lines 170-175)
The literature indicated (Rozental et al., 2016) does not seem to be sufficient and adequate to justify this procedure. I am not aware of technical works on psychometrics that guide this type of data treatment even before submitting the items to exploratory factor analysis – these multivariate analyses. In my assessment it makes no sense to make decisions about excluding items before assessing their multivariate association pattern in an exploratory factor analysis (EFA).
Authors: We absolutely agree with this important point, highlighting that our approach was unclearly explained, as we previously answered in the comment referred to in the abstract. In the following sentences, there are our edits and additional literature to support our approach.
“Before performing the EFA in the first group of responders, an item analysis via corrected item-total correlations was computed to preliminary select those items that better supported a stable assessment of the underlying measurement. Therefore, the analysis of corrected item-total correlations helped to reduce items whose elimination improved reliability coefficient alpha to the ones that supported the highest reliability of the scale for determining the correlation matrix for the EFA, also avoiding the risk of multicollinearity (Rozental et al., 2016). Although this preliminary analysis is not required to strictly create an adequate squared matrix for factor analyses because there is an availability of robust estimation methods, items with poor item-total correlations could compromise the scale’s validity, consequently reducing its capacity to identify a stable construct in the Italian-speaking settings (Hinkin, 1995). Owing to this possibility to reduce the items before the validity test as per previous studies (Cristobal et al., 2007; Hinkin, 1995) and, acknowledging possible benefits derived from having a briefer scale with higher internal consistency, the items with an item-total correlation < .30 or > .90 were removed for their effects on the reliability before the validity tests (Cristobal et al., 2007)”.
Cristobal, E., Flavián, C., & Guinalíu, M. (2007). Perceived e‐service quality (PeSQ): Measurement validation and effects on consumer satisfaction and web site loyalty. Managing Service Quality: An International Journal, 17(3), 317–340. https://doi.org/10.1108/09604520710744326
Hinkin, T. R. (1995). A Review of Scale Development Practices in the Study of Organizations. Journal of Management, 21(5), 967–988. https://doi.org/10.1177/014920639502100509
Rozental, A., Kottorp, A., Boettcher, J., Andersson, G., & Carlbring, P. (2016). Negative Effects of Psychological Treatments: An Exploratory Factor Analysis of the Negative Effects Questionnaire for Monitoring and Reporting Adverse and Unwanted Events. PloS One, 11(6), e0157503. https://doi.org/10.1371/journal.pone.0157503
What is indicated is to verify a series of assumptions that the EFA requires, such as: presence and systematicity of outliers (there is no report in the text that was evaluated); multivariate normality (there is no report in the text that was evaluated); choice of estimators suitable for the measurement level (assumption that the study ignores when using Pearson matrices and more suitable estimators for interval data, when the RIPLS measurement level is ordinal, for which polychoric correlation matrices are the most indicated in analyzes factorials), multicollinearity (assuming that the study evaluates in a simplistic way and, in my opinion, wrong when it comes to preparing the database for an EFA); linearity (there is no report in the text that was evaluated). Having verified these assumptions, the assessment of the retention or not of items must be based on the parameters estimated by the EFA, for example, the saturation of the item in the factor; the item-total correlation of the items in their respective factors and, obviously, the theoretical-interpretative plausibility.
Authors: We fully agree to report the tests for assessing the assumptions that EFA requires in the manuscript. In the current version, we added the required information, the details of the references and the SPSS macros that we used.
“After removing uncorrelated items, the assumptions underpinning the possibility of performing an Exploratory Factor Analysis (EFA) were tested: systematic outliers were assessed using a visual inspection based on the observed univariate distribution and removed [38], linearity between items was assessed using bivariate scatterplots [39], and the Mardia's test for multivariate normality was performed in Amos environment [40]. In addition, as the level of measurement for each item was ordinal, a polychoric correlation matrix was used instead of the default Pearson’s matrix by employing the macro developed by Basto and Pereira [41].”
In our case, the issue of multicollinearity was managed by the employed approach to reduce items exploring the item-to-total correlations before EFA as per our previous explanations.
The authors make this decision to exclude items based on bivariate analyzes and not on multivariate models capable of evaluating how well these items are grouped together with the others – element evaluated after the EFA, not before; assume unidimensionality of the set of items when calculating the item-total correlation.
Authors: We really appreciated this comment as it allowed us to discuss our approach further. As previously stated, some misunderstandings may result from reading the wording on the updated item-total correlations. The explanation of our approach in the prior version was actually misleading and we acknowledge that this approach is based on our willingness to keep for EFA only those items that concur to improve internal consistency as stated in previous studies:
- Hinkin TR. A Review of Scale Development Practices in the Study of Organizations. Journal of Management. 1995;21(5):967-988. doi:10.1177/014920639502100509 [on page 975]
- Parasuraman, A., Zeithaml, V. A., & Malhotra, A. (2005). ES-QUAL: A multiple-item scale for assessing electronic service quality. Journal of Service Research, 7(3), 213-233. [on page 219]
- Barrera, R. B., García, A. N., & Moreno, M. R. (2014). Evaluation of the e-service quality in service encounters with incidents: Differences according to the socio-demographic profile of the online consumer. Revista Europea de Dirección y Economía de la Empresa, 23(4), 184-193.
- Cristobal, E., Flavián, C., & Guinalíu, M. (2007). Perceived e‐service quality (PeSQ): Measurement validation and effects on consumer satisfaction and web site loyalty. Managing Service Quality: An International Journal, 17(3), 317–340. https://doi.org/10.1108/09604520710744326
- Zijlmans EAO, Tijmstra J, van der Ark LA, Sijtsma K. (2019). Item-Score Reliability as a Selection Tool in Test Construction. Front Psychol. 11;9:2298. doi: 10.3389/fpsyg.2018.02298
We also added the following paragraph in the limitations:
“Another relevant aspect that might play as a limitation is the decision to preliminary remove from the scale the items showing non-significant corrected item-total correlations. Even if this approach is consistent with previous research [36,37], we have to acknowledge that the availability of robust methods for estimates of multivariate analyses would have been adequate to manage an initial validation assessment before performing the reliability assessments. Also, an exploratory factor analysis would have been adequate to address the possible deletion of ambiguous items. We preferred to perform a preliminary reliability assessment to optimize the subsequent psychometric evaluations of the items following the simulation study of Zijlmans and colleagues that supported the idea of performing a corrected item-total correlation to omit items from the preliminary tests before validity assessments [52].”
Another problem that I identify in the methodological choices concerns the use of the K1 / scree plot criterion to choose the number of factors to be retained. This criterion has been heavily criticized in the psychometric literature for being subjective and ambiguous (see list below of works that can help). One study of psychometric simulation, in particular, demonstrated that the parallel analysis criterion (Horn, 1965) is one of the most accurate, indicating the correct number of factors in the matrix in 92% of the cases involved in the simulations – with the scree test, only 52% accuracy (Zwick & Velicer, 1986).
Anyway, in my opinion, this decision by the authors is wrong and compromises the entire list of results presented in the sequence.
Asún, R. A., Rdz-Navarro, K., & Alvarado,J. M. (2016). Developing multidimensional Likert scales using item factor analysis: The case of four-point items. Sociological Methods & Research, 45(1), 109-133. https://doi.org/10.1177/0049124114566716
Fabrigar, L. R., Wegener, D. T., MacCallum, R. C. & Strahan, E. J. (1999). Evaluating the use of exploratory factor analysis in psychological research. Psychological Methods, 4(3),272-299.
Hair, J.F.; Anderson, R.E.; Tatham, R.L.; Black, W.C. (2005). Multivariate Data Ananalysis, 6th ed., Prentice Hall, Englewood Cliffs, NJ
Hayton, J. C., Allen, D. G. & Scarpello, V. (2004). Factor retention decisions in exploratory factor analysis: A tutorial on parallel analysis. Organizational Research Methods, 7(2), 191-207.
Holgado–Tello, F. P., Chacón–Moscoso, S., Barbero–García, I., & Vila–Abad, E. (2010). Polychoric versus Pearson correlations in exploratory and confirmatory factor analysis of ordinal variables. Quality & Quantity, 44(1), 153.
Horn, J. L. (1965). A rationale and technique for estimating the number of factors in factor analysis. Psychometrika, 30(2), 179-185. https://doi.org/10.1007/BF02289447
Kline, R. B. (2011). Principles and practice of Structural Equation Modeling. 3th. The Guilford Press: New York, London.
Tabachnick, B. G., & Fidell, L. S. (2013). Using multivariate statistics (6th ed.) New Jersey: Pearson Education.
Zwick, W. R. & Velicer, W. F. (1986). Comparison of Five Rules of Determining the Number of Componentes to Retain. Multivariate Behavioral Research, 17, 258-269.
Authors: We fully agree with you; for this reason, we have integrated a parallel analysis into our study. The number of factors could be accurately determined using this method, based on simulated random data to estimate the number of factors. Using a Monte Carlo Simulation, a random simulative (artificial) data set is therefore generated besides the actual (real) data set, and the estimated eigenvalues are calculated as per the literature indicated in your comment. When the method is employed, the number of factors where the eigenvalue in the simulative sample is higher than that of the actual data is considered significant.
Literature used to address our interpretation of the parallel analysis:
- Silverstein, A. B. (1977). Comparison of two criteria for determining the number of factors. Psychological Reports, 41, 387–390.
- Silverstein, A. B. (1987). Note on the parallel analysis criterion for determining the number of common factor or principal components. Psychological Reports, 61, 351–354.
We adapted the macro described by O’Connor (2000) in to a specific syntax for our study to run a parallel analysis
O'Connor, B. P. (2000). SPSS and SAS programs for determining the number of components using parallel analysis and Velicer's MAP test. Behavior Research Methods, Instrumentation, and Computers, 32, 396-402.
Encoding: UTF-8.
* Parallel Analysis program.
set mxloops=9000 printback=off width=80 seed = 1953125.
matrix.
* enter your specifications here.
compute ncases = 111.
compute nvars = 14.
compute ndatsets = 100.
compute percent = 95.
* Specify the desired kind of parallel analysis, where:
1 = principal components analysis
2 = principal axis/common factor analysis.
compute kind = ??? .
****************** End of user specifications. ******************
* principal components analysis.
do if (kind = 1).
compute evals = make(nvars,ndatsets,-9999).
compute nm1 = 1 / (ncases-1).
loop #nds = 1 to ndatsets.
compute x = sqrt(2 * (ln(uniform(ncases,nvars)) * -1) ) &*
cos(6.283185 * uniform(ncases,nvars) ).
compute vcv = nm1 * (sscp(x) - ((t(csum(x))*csum(x))/ncases)).
compute d = inv(mdiag(sqrt(diag(vcv)))).
compute evals(:,#nds) = eval(d * vcv * d).
end loop.
end if.
* principal axis / common factor analysis with SMCs on the diagonal.
do if (kind = 2).
compute evals = make(nvars,ndatsets,-9999).
compute nm1 = 1 / (ncases-1).
loop #nds = 1 to ndatsets.
compute x = sqrt(2 * (ln(uniform(ncases,nvars)) * -1) ) &*
cos(6.283185 * uniform(ncases,nvars) ).
compute vcv = nm1 * (sscp(x) - ((t(csum(x))*csum(x))/ncases)).
compute d = inv(mdiag(sqrt(diag(vcv)))).
compute r = d * vcv * d.
compute smc = 1 - (1 &/ diag(inv(r)) ).
call setdiag(r,smc).
compute evals(:,#nds) = eval(r).
end loop.
end if.
* identifying the eigenvalues corresponding to the desired percentile.
compute num = rnd((percent*ndatsets)/100).
compute results = { t(1:nvars), t(1:nvars), t(1:nvars) }.
loop #root = 1 to nvars.
compute ranks = rnkorder(evals(#root,:)).
loop #col = 1 to ndatsets.
do if (ranks(1,#col) = num).
compute results(#root,3) = evals(#root,#col).
break.
end if.
end loop.
end loop.
compute results(:,2) = rsum(evals) / ndatsets.
print /title="PARALLEL ANALYSIS:".
do if (kind = 1).
print /title="Principal Components".
else if (kind = 2).
print /title="Principal Axis / Common Factor Analysis".
end if.
compute specifs = {ncases; nvars; ndatsets; percent}.
print specifs /title="Specifications for this Run:"
/rlabels="Ncases" "Nvars" "Ndatsets" "Percent".
print results /title="Random Data Eigenvalues"
/clabels="Root" "Means" "Prcntyle" /format "f12.6".
do if (kind = 2).
print / space = 1.
print /title="Compare the random data eigenvalues to the".
print /title="real-data eigenvalues that are obtained from a".
print /title="Common Factor Analysis in which the # of factors".
print /title="extracted equals the # of variables/items, and the".
print /title="number of iterations is fixed at zero;".
print /title="To obtain these real-data values using SPSS, see the".
print /title="sample commands at the end of the parallel.sps program,".
print /title="or use the rawpar.sps program.".
print / space = 1.
print /title="Warning: Parallel analyses of adjusted correlation matrices".
print /title="eg, with SMCs on the diagonal, tend to indicate more factors".
print /title="than warranted (Buja, A., & Eyuboglu, N., 1992, Remarks on parallel".
print /title="analysis. Multivariate Behavioral Research, 27, 509-540.).".
print /title="The eigenvalues for trivial, negligible factors in the real".
print /title="data commonly surpass corresponding random data eigenvalues".
print /title="for the same roots. The eigenvalues from parallel analyses".
print /title="can be used to determine the real data eigenvalues that are".
print /title="beyond chance, but additional procedures should then be used".
print /title="to trim trivial factors.".
print / space = 1.
print /title="Principal components eigenvalues are often used to determine".
print /title="the number of common factors. This is the default in most".
print /title="statistical software packages, and it is the primary practice".
print /title="in the literature. It is also the method used by many factor".
print /title="analysis experts, including Cattell, who often examined".
print /title="principal components eigenvalues in his scree plots to determine".
print /title="the number of common factors. But others believe this common".
print /title="practice is wrong. Principal components eigenvalues are based".
print /title="on all of the variance in correlation matrices, including both".
print /title="the variance that is shared among variables and the variances".
print /title="that are unique to the variables. In contrast, principal".
print /title="axis eigenvalues are based solely on the shared variance".
print /title="among the variables. The two procedures are qualitatively".
print /title="different. Some therefore claim that the eigenvalues from one".
print /title="extraction method should not be used to determine".
print /title="the number of factors for the other extraction method.".
print /title="The issue remains neglected and unsettled.".
end if.
end matrix.
* Commands for obtaining the necessary real-data eigenvalues for
principal axis / common factor analysis using SPSS;
make sure to insert valid filenames/locations, and
remove the '*' from the first columns.
* correlations var1 to var20 / matrix out ('filename') / missing = listwise.
* matrix.
* MGET /type= corr /file='filename' .
* compute smc = 1 - (1 &/ diag(inv(cr)) ).
* call setdiag(cr,smc).
* compute evals = eval(cr).
* print { t(1:nrow(cr)) , evals }
/title="Raw Data Eigenvalues"
/clabels="Root" "Eigen." /format "f12.6".
* end matrix.
Following Turner’s indications, our results performing the parallel analysis confirmed the plausible two-factor structure of the 14-item scale.
Turner, N. E. (1998). The effect of common variance and structure pattern on random
data eigenvalues: Implications for the accuracy of parallel analysis. Educational
and Psychological Measurement, 58(4), 541-568. doi: http://dx.doi.org/10.1177/0013164498058004001
PARALLEL ANALYSIS:
Specifications for this Run:
Ncases 111
Nvars 14
Ndatsets 100
Percent 95
Random Data Eigenvalues
Root Means Prcntyle
1,000000 1,655881 1,823377
2,000000 1,491044 1,576016
3,000000 1,366219 1,411775
4,000000 1,258741 1,335097
5,000000 1,172291 1,241021
6,000000 1,083630 1,150413
7,000000 1,000473 1,065226
8,000000 ,924056 ,994991
9,000000 ,857359 ,915420
10,000000 ,783006 ,843926
11,000000 ,717451 ,777307
12,000000 ,645699 ,699438
13,000000 ,566267 ,627911
14,000000 ,477884 ,550303
------ END MATRIX -----
Line 194: “Adequate fit indexes were: CFI ≥ 0.90, TLI ≥ 0.90, RMSEA lower than 0.80, and SRMR lower than 0.1.”
It remains to refer to the literature that supports these criteria. In Kline (2011), for example, values are accepted for RMSEA < 0.08 and for SRMR < 0.06.
Authors: We added the reference used to support our statement, which precisely states that RMSEA values less than 0.05 are good, values between 0.05 and 0.08 are acceptable, values between 0.08 and 0.1 are marginal, and values greater than 0.1 are poor. Fabrigar, L. R., Wegener, D. T., MacCallum, R. C., & Strahan, E. J. (1999). Evaluating the use of exploratory factor analysis in psychological research. Psychological Methods, 4(3), 272–299. https://doi.org/10.1037/1082-989X.4.3.272 ]
RESULTS
Table 3: It makes no sense to assume, before the dimensionality analysis/factor analysis, that the items are one-dimensional (to calculate Cronbach's Alpha this assumption must have been verified before!). If there is already robust evidence of this assumption of one-dimensionality of these items in the respective factors in the literature, why not start with a confirmatory approach? Exploratory analysis is used precisely to evaluate these groupings, excluding items before seems to me to be wrong, as explained above. Furthermore, a total Cronbach's alpha makes no sense because the scale does not have an overall score – this total score has never been modeled in the studies I know of with RIPLS, in any of its versions. There are also problems in assuming the Likert scale as a measure at an interval level, when in fact this scale measures at an ordered ordinal/categorical level.
Authors: Thank you for this comment; very well appreciated. We hope that the edits and the explanations provided in answering your useful comments on the methods are adequate to clarify our approach.
We also added the following paragraph in the limitations:
“Another relevant aspect that might play as a limitation is the decision to preliminary remove from the scale the items showing non-significant corrected item-total correlations. Even if this approach is consistent with previous research [36,37], we have to acknowledge that the availability of robust methods for estimates of multivariate analyses would have been adequate to manage an initial validation assessment before performing the reliability assessments and exploratory factor analyses would have been adequate to address the possible deletion of ambiguous items. We preferred to perform a preliminary reliability assessment to optimize the subsequent psychometric evaluations of the items following the simulation study of Zijlmans and colleagues that supported the idea of performing a corrected item-total correlation to omit items from the preliminary tests before validity assessments [52].”
Lines 238-239: “The most suitable solution from the EFA interpreting the scree plot and extracting factors with eigenvalues greater than 1.0 was a two-factor solution.”
As already mentioned, the use of this criterion is widely criticized in the psychometric literature. As the empirical eigenvalues obtained in the study were not presented, this reviewer is unable to perform the parallel analysis and better evaluate the results of the study. It is suggested that the eigenvalues obtained from the data matrix be presented.
Authors: Thank you for this precious comment. We added the corroboration of the method utilized by employing a parallel analysis as per your suggestion.
Figure 2 and Lines 253-255: “However, by exploring possible specifications to the model, the residuals of item 4 and item 10 have been correlated by accounting for the modification index and the wording of the two items.”
By correlating errors, the authors again start with an exploratory approach. They misrepresent the true nature of AFC, falling back into an exploratory approach. This is because, by including correlation arrows between the errors, the chi-square value is reduced and all the model's adequacy indices are improved, so nothing is being “tested”. By including arrows “suggested” by the modification indices, confirmatory analysis is left aside, returning to an exploratory approach. For each arrow there should be a theoretical justification for its and, later, a new analysis with a validation sample should be performed. I suggest consulting:
Breckler, S. J. (1990). Applications of Covariance Structure Modeling in Psychology: Cause for Concern? Psychological Bulletin,107(2), 260-273.
Authors: We agree with this fantastic comment and propose a justification of our approach that we also integrated into the current version of the manuscript. This is the first validation study in the Italian-speaking population. As discussed, the need for coming back to an exploratory approach was given by the high likelihood of having inter-correlated errors for the similarity of the items. Therefore, the authors agreed that “for each arrow there should be a theoretical justification for its and, later, a new analysis with a validation sample should be performed,” and the theoretical justification is the Italian-translated wording of the items. To allow readership a full interpretation, in fact, we kept both models: specified and unspecified. The specification was fully consistent with the approach indicated by Whittaker (2012) and we added, in the limitation section, the need for further testing the validated tool in another sample.
Whittaker, T. A. (2012). Using the modification index and standardized expected parameter change for model modification. The Journal of Experimental Education, 80(1), 26-44.
“The model specification means that the confirmatory approach became more exploratory than the unspecified model; therefore, future corroborations in different samples are needed for defining the psychometric performance of this new version of the I-RIPLS encompassing 14 items and two factors”.
Table 5: There are inaccuracies. The item-total correlation should only be calculated between the items of each factor, not for the scale as a whole. Furthermore, as already explained, it makes no sense to calculate a Cronbach's alpha for the entire scale - this factor has not been modeled. It is not possible to calculate the internal consistency or even calculate the score without unequivocal evidence of the unidimensionality of the set of items under analysis.
Authors: We agree with your considerations. For this reason, we thought that the table could be criticized. We have removed the table, and the compute on Cronbach's alpha regarding the overall scale.
Table 6: In my opinion, the data presented in Table 6 are not the results of the analysis of data derived from the field research, but rather the literature review. Therefore, this Table does not seem to make sense here in the Results section – it would be better if it were presented in the Introduction of the manuscript.
Furthermore, I note that among the 20 studies reported, regardless of the versions/types of items, most propose three (n = 12) or four factors (n = 6) of readiness for IPE - only two studies proposed two factors to represent the structure of RIPLS. This fact, together with all the previous comments about methodological choices in psychometric analyses, lead me to more consistent assumptions about methodological inadequacies in the evaluation of the internal structure / dimensionality of the RIPLS presented in the manuscript.
Considering everything previously, my suggestion to redo the psychometric evaluations of using the most appropriate manuscript, according to the literature indicated throughout the evaluation and return for evaluation. Unless better evaluated, this is my opinion.
Authors: We agree. We have moved that table to the introduction and extensively revised our analytical approach. Thank you so much for your outstanding revision. We will be happy to provide more information and clarification concerning our manuscript.

Reviewer 4 Report
The authors provide an excellent description of the translation and validation process of the RIPLS. The writing was clear and concise, and the processes were easy to navigate. The authors should consider including some discussion regarding the translation process in the discussion (ie., limitations). Were there any issues that arose during the process?
Further, additional information regarding the removal of the Negative professional identity and Roles and responsibilities in the discussion is warranted. How does this change the instrument from the original english?
The description of the study limitations is appreciated. However, is the reader to assume that all participants were fluent in Italian since it is the official language of the institution? Were all participants native Italian speakers or was this their second language?
Finally, what does Table 6 add to the results and discussion? This either needs to be developed further or removed.
Author Response
Thank you immensely for the opportunity to continue to develop our paper. We appreciate the valuable comments and suggestions provided, and we have considered all the points and have revised our manuscript according to them. We will be happy to provide more information and clarification concerning our manuscript if needed.
The authors provide an excellent description of the translation and validation process of the RIPLS. The writing was clear and concise, and the processes were easy to navigate. The authors should consider including some discussion regarding the translation process in the discussion (ie., limitations). Were there any issues that arose during the process?
Authors: Thank you for this comment; we edited the limitations as per your suggestion.
Further, additional information regarding the removal of the Negative professional identity and Roles and responsibilities in the discussion is warranted. How does this change the instrument from the original English?
Authors: Thank you for this comment. We added information and debate regarding the removed items in the current version, as also underlined by reviewer #3.
The description of the study limitations is appreciated. However, is the reader to assume that all participants were fluent in Italian since it is the official language of the institution? Were all participants native Italian speakers or was this their second language?
Authors: All the students had a high linguistic competence and fluency in speaking and writing Italian as the university is an Italian university abroad, and all the activities are performed in Italian. However, we agree with your comment that the fact that some students were not native speakers has to be acknowledged in the limitations.
Finally, what does Table 6 add to the results and discussion? This either needs to be developed further or removed.
Authors: Thank you for this very useful comment. As also suggested by other reviewers #1 and #3, we have moved this table to the background.

Round 2
Reviewer 3 Report
Dear authors and editors, Although I have the option to exclude the items before the discord from the analysis to continue the analysis and to reference the best text in the text, the study was successfully carried out / the strategy adopted for the analysis. Other proposals presented were presented, so I consider the revised version presented and can be indicated for publication. Only one last correction must be made. The literature recommends values lower than 0.08 for the RMSEA index and not lower than 0.80, as stated in line 228 of the revised version of the manuscript. Unless better evaluated, this is my opinion.Author Response
Dear authors and editors, Although I have the option to exclude the items before the discord from the analysis to continue the analysis and to reference the best text in the text, the study was successfully carried out / the strategy adopted for the analysis. Other proposals presented were presented, so I consider the revised version presented and can be indicated for publication. Only one last correction must be made. The literature recommends values lower than 0.08 for the RMSEA index and not lower than 0.80, as stated in line 228 of the revised version of the manuscript. Unless better evaluated, this is my opinion.
Authors: Dear reviewer, thanks a lot for your efforts in reviewing our work; we really appreciate it. We have corrected the typo in the current version.
